# Queried Unlabeled Data Improves and Robustifies Class-Incremental Learning

**Tianlong Chen**                                                    *tianlong.chen@utexas.edu*
*University of Texas at Austin*

**Sijia Liu**                                                                *liusiji5@msu.edu*
*Michigan State University*
*MIT-IBM Watson AI Lab, IBM Research*

**Shiyu Chang**                                                            *chang87@ucsb.edu*
*University of California, Santa Barbara*

**Lisa Amini**                                                          *lisa.amini@us.ibm.com*
*MIT-IBM Watson AI Lab, IBM Research*

**Zhangyang Wang**                                                      *atlaswang@utexas.edu*
*University of Texas at Austin*

**Reviewed on OpenReview:** *https://openreview.net/forum?id=oLvlPJheCD*

## Abstract

Class-incremental learning (CIL) suffers from the notorious dilemma between learning newly added classes and preserving previously learned class knowledge. That catastrophic forgetting issue could be mitigated by storing historical data for replay, which yet would cause memory overheads as well as imbalanced prediction updates. To address this dilemma, we propose to leverage "free" external unlabeled data querying in continual learning. We first present a CIL with Queried Unlabeled Data (**CIL-QUD**) scheme, where we only store a handful of past training samples as anchors and use them to query relevant unlabeled examples each time. Along with new and past stored data, the queried unlabeled are effectively utilized, through learning-without-forgetting (LwF) regularizers and class-balance training. Besides preserving model generalization over past and current tasks, we next study the problem of adversarial robustness for CIL-QUD. Inspired by the recent success of learning robust models with unlabeled data, we explore a new robustness-aware CIL setting, where the learned adversarial robustness has to resist forgetting and be transferred as new tasks come in continually. While existing options easily fail, we show queried unlabeled data can continue to benefit, and seamlessly extend CIL-QUD into its robustified versions, **RCIL-QUD**. Extensive experiments demonstrate that CIL-QUD achieves substantial accuracy gains on CIFAR-10 and CIFAR-100, compared to previous state-of-the-art CIL approaches. Moreover, RCIL-QUD establishes the first strong milestone for robustness-aware CIL. Codes are available in https://github.com/VITA-Group/CIL-QUD.

## 1 Introduction

Most deep neural networks (DNNs) are trained when the complete dataset and all class information are available at once and fixed. However, real-world applications, such as robotics and mobile health, often demand learning classifiers continually (Parisi et al., 2019), when the data and classes are presented and fitted sequentially. Such *continual learning* pose severely challenge for standard DNNs, where previous experiences easily get overwritten as more data and new tasks arrive, i.e., the notorious *catastrophic forgetting* (Goodfellow et al., 2013; McCloskey & Cohen, 1989).

This paper is focused on a realistic yet challenging setting of continual learning, called *class-incremental learning* (CIL) (Rebuffi et al., 2017; Belouadah & Popescu, 2019; 2020; Zhang et al., 2020). In CIL, the classifier model will need to be incrementally re-trained from time to time, when new classes are added. Ideally, the re-training should provide a competitive multi-class classifier for all classes observed so far at any time. Unfortunately, naively augmenting and fine-tuning the model to learn new classes will only see an abrupt degradation of performance on the original set of classes, when the training objective is adapted to the newly added set of classes. Several attempts have been made by storing past training data (Castro et al., 2018; Javed & Shafait, 2018; Rebuffi et al., 2017; Belouadah & Popescu, 2019;

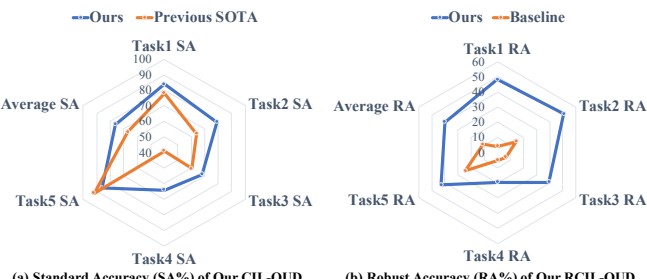

Figure 1: Summary of our achieved performance on CIFAR-10, where CIL is conducted over 5 incremental learning tasks (each 2-class). Figure (a) presents the standard accuracy (SA) achieved by CIL-QUD versus the previous SOTA (Belouadah & Popescu, 2019). Figure (b) shows the robust accuracy (RA) achieved by RCIL-QUD versus the baseline of directly applying adversarial training (Madry et al., 2018) to CIL.

2020), by generative models (Kemker & Kanan, 2018; Shin et al., 2017; Li et al., 2020), or by regularized fine-tuning (Aljundi et al., 2017; Li & Hoiem, 2017). However, they suffer from either excessive memory overhead or the so-called asymmetric information problem (Zhang et al., 2020), learning this CIL problem far from being resolved. More detailed discussions will be presented in Section 2.

We instead embrace another potential blessing: using publicly available unlabeled auxiliary data in CIL, which can be obtained at negligible costs (e.g., crawled from the web). The power of unlabeled data has recently drawn explosive interests, in multiple contexts such as semi-supervised learning (Chen et al., 2020d), self-training (Xie et al., 2020), and self-supervised learning (Chen et al., 2020e;c). With massive "free" unlabeled data serving pre-training or regularization, those prior works significantly reduced their reliance on labeled data, sometimes performing on par with fully supervised classification while using one to two magnitudes less labeled samples.

Particularly for CIL, the training data and labels are not only expensive to *collect* (same as standard learning), but also expensive to *store* after its own training task finishes. The memory cost can become predominant, and further in many cases the legacy data cannot remain accessible after training, due to legal or proprietary reasons. That naturally makes external unlabeled data even more promising for CIL: it is a cheap substitute for past training data and needs not to be always stored. Lately, Zhang et al. (2020) presented a new paradigm that proved the concept of improving CIL with external unlabeled data. Each time as new tasks arrive, the authors first trained a separate new model for the new classes using labeled data, and then distilled the two teacher models (the old and new models) into one student model. The second step leveraged unlabeled auxiliary data for distillation, which the authors claimed would help debias the knowledge transferred from both teacher models, compared to using either old or new training data. Their method delivered solid accuracies on CIL classification benchmarks. However, their proposed method was limited by the inefficiency of performing tedious two-step training each time (first the new teacher model, and then the distillation); and the distillation step uses all unlabeled data (∼1 million images in their default case), which further causes significant training burdens.

### 1.1 Our Contributions

We seek to push forward the utility and potential of unlabeled data in CIL, by asking two further questions:

**Q1:** *Provided with massive unlabeled data, can we sample and leverage them with higher efficiency?*
**Q2:** *From unlabeled data, can we harvest more "bonus" in other CIL performance dimensions besides accuracy?*

Our answer to **Q1** draws the best ideas from two worlds. On the one hand, existing methods that store and replay past training data (Castro et al., 2018; Javed & Shafait, 2018; Rebuffi et al., 2017; Belouadah & Popescu, 2019; 2020) are still the most effective in overcoming catastrophic forgetting, despite its storage headache and

sometimes the data privacy/copyright concerns. On the other hand, Zhang et al. (2020) pioneered on CIL with unlabeled data, but sacrificed training/data efficiency as above explained. To mitigate their respective challenges, we propose to integrate the complementary strengths of *storing past data* and *leveraging unlabeled data*; specifically, we only store a handful of historical samples, which would be used as *anchor points* to *query the most relevant unlabeled data*. Aided by the learning-without-forgetting (LwF) regularizer (Li & Hoiem, 2017), those queried unlabeled samples join the labeled data from the newly added class to balance between preserving the historical and learning the new knowledge. Our ablation experiments endorse that (1) such queried unlabeled data leads to state-of-the-art (SOTA) CIL performance while sacrificing no efficiency; and (2) using anchor-based query stably outperforms random selection, and stays robust to the unlabeled data distribution shifts and volume variations.

Our answer to **Q2** is strongly motivated by the prevailing success of utilizing unlabeled data besides standard classification, e.g., improving robustness (Alayrac et al., 2019). We hope to extend and validate those benefits to CIL as well. Specifically, adversarial robustness (Chen et al., 2020a) arises as a key demand when deploying DNNs to safety/security-critical applications. While continual learning has been so far focused on maintaining *accuracy* across the stream of tasks, we consider it the same necessary - if not more - to examine whether the model can maintain *robustness* across old and new tasks: a critical step towards enabling trustworthy learning in the open world. This problem has been unfortunately largely overlooked and under-explored in the CIL regime. In fact, even one-shot transferability (Hendrycks et al., 2019; Shafahi et al., 2020; Goldblum et al., 2020; Chan et al., 2019) of robustness (from one source domain pre-trained model, to one target domain new task) has not been studied until recently, and was shown to be challenging. We fill in this research gap, by (1) for the first time, studying the *catastrophic forgetting of robustness*[1] and showing that it cannot be trivially fixed; and (2) extending our new framework with unlabeled data query to sustaining strong robustness in the CIL scheme, with several robustified regularizations. That provides both an extra benchmark dimension for CIL, and a significant advance in the study of transferable robustness beyond one-shot.

Our specific contributions are summarized below:

- A novel framework of *Class-Incremental Learning with Queried Unlabeled Data* (**CIL-QUD**), that seamlessly integrates a handful of stored historical samples and unlabeled data through anchor-based query. CIL-QUD also carefully leveraged LwF regularizers (Li & Hoiem, 2017) and balanced training (Zhang et al., 2019b) as building blocks. It has light overheads in the model, data storage, and training.

- A first-of-its-kind study of preserving adversarial robustness in CIL, and an extension of CIL-QUD to its robustified version called **RCIL-QUD**. Specifically, we propose and compare two robust versions of LwF regularizers together with an add-on robust regularizer, which are built on the queried unlabeled data.

- Experiments demonstrating that the power of unlabeled data can effectively extend to CIL, contributing substantially to superior accuracy as well as adversarial robustness - both preserved without catastrophic forgetting. In Figure 1, CIL-QUD outperforms previous SOTA by large margins of 10.28% accuracy on CIFAR-10; on CIFAR-100, it also outperforms (Zhang et al., 2020) by a 1.19% margin. RCIL-QUD further establishes the first strong milestone for robustness-aware CIL, that significantly surpasses existing baselines.

## 2 Related Work

**Class-incremental Learning** Among numerous methods developed, one category of approaches (Wang et al., 2017; Rosenfeld & Tsotsos, 2018; Rusu et al., 2016; Aljundi et al., 2017; Rebuffi et al., 2018; Mallya et al., 2018) incrementally grow the model capacity to accommodate new classes, yet suffering from the explosive model size as well as inflexibility (e.g., requiring task ID at inference). Another category of solutions is based on transfer learning (Kemker & Kanan, 2018; Belouadah & Popescu, 2018), whose effectiveness yet hinges on the pre-training quality.

---

[1]It is similar to the catastrophic forgetting of standard generalization but with different evaluation metrics. In other words, robustifying models on a new task leads to robustness degradations on previous tasks, as shown in Figure 1 (*right*).

A popular and successful family of CIL methods (Li & Hoiem, 2017; Castro et al., 2018; Javed & Shafait, 2018; Rebuffi et al., 2017; Belouadah & Popescu, 2019; 2020) (partially) memorized past training data to fight catastrophic forgetting, bypassing the need for dynamic model capacity. Many of those algorithms viewed CIL as an imbalanced learning problem, where previous and newly added classes made an class size extreme disparity (He & Garcia, 2009; Buda et al., 2018). Learning without Forgetting (*LwF*) (Li & Hoiem, 2017) made an attempt to fix that dilemma through an LwF regularization via knowledge distillation. Although the original *LwF* did not deposit past data, later works (Rebuffi et al., 2017) augmented it with a memory bank of previous tasks and previously stored data. More follow-ups (Castro et al., 2018; He et al., 2018; Javed & Shafait, 2018; Rebuffi et al., 2017; Belouadah & Popescu, 2019; 2020) advanced the LwF regularization further. For example, *IL2M* (Belouadah & Popescu, 2019) utilized a second memory bank to store past class statistics obtained at past training, and incorporated those stored statistics to compensate for the previous (minority) classes' predictions.

Since storing past data inevitably incurred memory overheads, He et al. (2018) used GANs to generate exemplars for previous tasks, for supplying a re-balanced training set at each incremental state. The most relevant work to ours is Zhang et al. (2020), which pioneered the usage of free external unlabeled data to boost CIL, as we have discussed in Section 1. In comparison, **our proposed CIL-QUD clearly distinguishes itself** from Zhang et al. (2020), by (1) using adaptively queried unlabeled samples based on stored anchors, instead of full unlabeled data; (2) avoiding two-stage training and instead adopting LwF-regularized fine-tuning; (3) being the first in CIL to combine training with class-balanced and randomly sampled data (Zhang et al., 2019b). Besides, neither Zhang et al. (2020) nor any other mentioned work ever touched the robustness preservation in CIL. More details on those differences are expanded in Section 3.

On the other hand, recent investigations (Wang et al., 2021; He & Zhu, 2021) also explore other alternative possibilities of leveraging unlabelled data to boost continual learning. For example, Wang et al. (2021) replays unlabeled data sampled from a conditional generator, and utilizes a consistency regularization to learn an improved continual classifier. He & Zhu (2021) studies continual learning in a fully unsupervised mode by assigning unlabeled data with clustered pseudo labels. Meanwhile, Bateni et al. (2022) and Chen et al. (2020b) enable few-shot and efficient continual learning respectively, with the assistance of unlabeled data. A recent survey paper (Qu et al., 2021) also provides a good summary of current achievements in this field.

**Adversarial Robustness and Its Transferablity** DNNs commonly suffer from adversarial vulnerability (Goodfellow et al., 2014), and numerous defense methods have been invented (Madry et al., 2018; Sinha et al., 2018; Rony et al., 2019; Zhang et al., 2019a; Ding et al., 2020). However, most of them focus on attacking/defending DNNs trained on a single fixed dataset and task. Recently, a couple of works emerge to cogitate the transferability of adversarial robustness (Hendrycks et al., 2019; Shafahi et al., 2020; Goldblum et al., 2020; Chan et al., 2019; Chen et al., 2020a), from a robust model pre-trained on a source domain to another target domain. They revealed that directly fine-tuning on the target domain data will quickly overwrite the pre-trained robustness (Chen et al., 2020a). One needs to refer to either adversarial fine-tuning on the target domain (Chen et al., 2020a), or specific regularizations such as knowledge distillation or gradient matching to preserve the source domain robustness knowledge (Shafahi et al., 2020; Goldblum et al., 2020; Chan et al., 2019). In comparison, we target at inheriting robustness across many sets of new data and tasks added continually, where the conventional transfer learning could be viewed as its oversimplified case. To our best knowledge, **our proposed RCIL-QUD marks the first-ever effort to explore this new daunting setting**.

## 3  Class-Incremental Learning with Queried Unlabeled Data (CIL-QUD)

In this section, we begin by presenting a brief background on class-incremental learning (CIL), and show our CIL-QUD that overcomes catastrophic forgetting by adaptively querying and leveraging unlabeled data.

### 3.1  CIL Preliminaries and Setups

As in Figure 2, CIL models are continuously trained over a *sequential data stream*, where a new classification task (consisting of *unseen classes*) could be added every time. This makes CIL highly challenging in contrast

to static learning and conventional (one-shot) transfer learning. Following Castro et al. (2018); He et al. (2018); Rebuffi et al. (2017), we consider a practical and challenging CIL setting with only a small memory bank $\mathcal{S}$ to store data from previous classes. Neither task ID nor order is pre-assumed to be known.

Let $\mathcal{T}_1, \mathcal{T}_2, \cdots$ represent a sequence of CIL tasks, and the $i$th task $\mathcal{T}_i$ contains data that fall in $(k_i - k_{i-1})$ classes $\mathcal{C}_i = \{c_{k_{i-1}+1}, c_{k_{i-1}+2}, \cdots, c_{k_i}\}$, with $k_0 = 1$ by convention. At the $i_{\text{th}}$ incremental learning session, we only have access to the training data associated with $\mathcal{T}_i$, and a limited number of stored data in $\mathcal{S}$ from the previous learning tasks. Let $f(\boldsymbol{\theta}, \boldsymbol{\theta}_{\text{c}}^{(i)}, \mathbf{x})$ denotes the mapping from input samples $\mathbf{x} \in \bigcup_{j=1}^{i} \mathcal{T}_j$ to the corresponding classes $\bigcup_{j=1}^{i} \mathcal{C}_j$ acquired from $\mathcal{T}_1, \cdots, \mathcal{T}_i$. Here $(\boldsymbol{\theta}, \boldsymbol{\theta}_{\text{c}}^{(i)})$ denotes the parameters of a CIL model updated

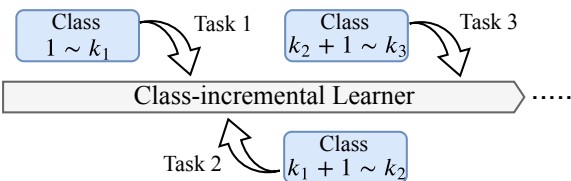

Figure 2: CIL learns continuously from a sequential data stream where new tasks are added. Each new task $i$ may contain several classes from class $k_{i-1} + 1$ to $k_i$. The CIL learner aims for multi-class classification for both previous and newly added classes.

till task $\mathcal{T}_i$. To be more specific, the CIL model consists of a feature extractor $\varphi(\boldsymbol{\theta}, \mathbf{x}) \in \mathbb{R}^d$ with parameters $\boldsymbol{\theta}$, which maps the input $\mathbf{x} \in \bigcup_{j=1}^{i} \mathcal{T}_j$ to a $d$-dimensional feature space. It is then followed by a multi-class (i.e., $\bigcup_{j=1}^{i} \mathcal{C}_j$) classifier with parameters $\boldsymbol{\theta}_{\text{c}}^{(i)}$, which maps the $d$-dimensional feature to the prediction vector $\rho_i(\boldsymbol{\theta}, \boldsymbol{\theta}_{\text{c}}^{(i)}, \mathbf{x})$ at the current task $\mathcal{T}_i$.

**Problem statement.** We now formally define our target CIL problem: Given a previously trained model $f(\hat{\boldsymbol{\theta}}, \hat{\boldsymbol{\theta}}_{\text{c}}^{(i-1)}, \mathbf{x})$ under tasks $\mathcal{T}_1, \cdots, \mathcal{T}_{i-1}$, the objective is to obtain an updated model $f(\boldsymbol{\theta}, \boldsymbol{\theta}_{\text{c}}^{(i)}, \mathbf{x})$ to preserve the generalization ability and robustness among all learned tasks, even if one can only have the access to data from the memory bank $\mathcal{S}$ as well as the current task $\mathcal{T}_i$. Here the feature extractor $\varphi(\boldsymbol{\theta}, \mathbf{x})$ is shared over all seen tasks, but the scale of the multi-class classifier linearly increases along with the incremental classes.

## 3.2 Anchor-based Query of Unlabeled Data

Catastrophic forgetting poses the major challenge to CIL. A major reason of forgetting is the asymmetric information between previous classes and newly added classes at each incremental learning stage. Existing CIL approaches (He & Garcia, 2009; Buda et al., 2018; Rebuffi et al., 2017; Belouadah & Popescu, 2019; 2020; Chu et al., 2016; Kemker & Kanan, 2018; Parisi et al., 2019) undertake the forgetting dilemma by training on new task data plus stored previous data. However, the inevitable storage limitation could still cause severe prediction biases as more classes come in. Chu et al. (2016); Kemker & Kanan (2018); Parisi et al. (2019) introduced balanced fine-tuning, yet incurring the risk of over-fitting new/minority classes. Zhang et al. (2020) tried to fix the dilemma by referring external unlabeled data. They did not store any past training data; but as above discussed suffered from considerable training overhead.

We present our novel remedy of utilizing anchor-based unlabeled data query, to balance between preserving the historical and learning the new knowledge. First, we store a small number of i.i.d randomly picked samples as "anchors" for every past training class, e.g., as few as ten samples per class on CIFAR-100. Then, during the next incremental processes, we query more auxiliary unlabeled samples from public sources with stored anchors, using certain similarity matches (see Section 3.4 for details). The queried samples are expected to present "similar" and more relevant information to past training data, compared to random samples. Practically, unlabeled samples can be queried from public sources containing diverse enough natural images, e.g., Google Images, that are not necessarily tied with previous classes. Next, we inject the previous information into learning new classes, by tuning with the learning-without-forgetting (LwF) regularizers (Li & Hoiem, 2017) on queried unlabeled data.

We next detail on the concrete regularizers $\mathcal{L}_{\text{LwF}}$ used. Let $\mathcal{U}$ donates the queried unlabeled data, $\mathcal{L}_{\text{LwF}}$ can be chosen from either knowledge distillation ($\mathcal{KD}$[2]) (Hinton et al., 2015; Li & Hoiem, 2017) or feature transferring ($\mathcal{FT}$[1]) (Shafahi et al., 2020), i.e., $\mathcal{L}_{\text{LwF}} \in \{\mathcal{KD}, \mathcal{FT}\}$.

---

[2]$\mathcal{KD}$ here denotes the regularization function of knowledge distillation, a modified cross-entropy as in Li & Hoiem (2017); $\mathcal{FT}$ denotes the regularization function of feature transferring, an $\ell_p$ distance metric as in Shafahi et al. (2020).

$\mathcal{KD}$ is one of the most classical regularizers in CIL (Li & Hoiem, 2017; Castro et al., 2018; He et al., 2018; Javed & Shafait, 2018; Rebuffi et al., 2017; Belouadah & Popescu, 2019; 2020), but was usually applied on fully-stored previous data and/or newly added data. In our case, we enforce the output probabilities $\rho(\boldsymbol{\theta}, \boldsymbol{\theta}_\mathrm{c}, \mathbf{x})$ of each queried unlabeled image $\mathbf{x} \in \mathcal{U}$ to be close to the recorded $\rho(\hat{\boldsymbol{\theta}}, \hat{\boldsymbol{\theta}}_\mathrm{c}, \mathbf{x})$. The $\mathcal{KD}$ regularization is then given by:

$$\mathcal{L}_{\mathrm{LwF}}(\boldsymbol{\theta}, \boldsymbol{\theta}_\mathrm{c}) := \mathbb{E}_{\mathbf{x} \in \mathcal{U}} \left[ \mathcal{KD} \left( \rho(\boldsymbol{\theta}, \boldsymbol{\theta}_\mathrm{c}, \mathbf{x}), \rho(\hat{\boldsymbol{\theta}}, \hat{\boldsymbol{\theta}}_\mathrm{c}, \mathbf{x}) \right) \right] \tag{1}$$

where $\boldsymbol{\theta}$ and $\boldsymbol{\theta}_\mathrm{c}$ present the parameters of current feature extractor and classifiers respectively, and $\hat{\boldsymbol{\theta}}$ and $\hat{\boldsymbol{\theta}}_\mathrm{c}$ stand for the previous ones.

$\mathcal{FT}$ (Shafahi et al., 2020) inherits previous knowledge by maximizing the similarity between current feature representations $\varphi(\boldsymbol{\theta}, \mathbf{x})$ and previous features $\varphi(\hat{\boldsymbol{\theta}}, \mathbf{x})$ on unlabeled data $\mathbf{x} \in \mathcal{U}$:

$$\mathcal{L}_{\mathrm{LwF}}(\boldsymbol{\theta}) := \mathbb{E}_{\mathbf{x} \in \mathcal{U}} \left[ \mathcal{FT}(\varphi(\boldsymbol{\theta}, \mathbf{x}), \varphi(\hat{\boldsymbol{\theta}}, \mathbf{x})) \right], \tag{2}$$

where $\varphi(\cdot)$ is the feature extractor, $\boldsymbol{\theta}$ is defined the same as above, and $\mathcal{FT}$ is a distance metric, which we choose to be $\ell_1$ norm here. An *interesting finding* from our later experiments (Table 1 and 2) is that, $\mathcal{KD}$ outperforms $\mathcal{FT}$ in preserving generalization ability under standard CIL, but $\mathcal{FT}$ becomes more useful in the later robustness-aware CIL.

### 3.3 Balanced Training with Auxiliary Classifiers

Another well-known remedy for the imbalanced classes is to re-sample mini-batches (Chawla et al., 2002; Haixiang et al., 2017; Tahir et al., 2009; Wang et al., 2019; Zhang et al., 2019b). It artificially re-balances between majority classes and minority classes to alleviate the negative effect of skewed training data distributions. In this paper, spurred by the recent progress in long-tail visual recognition (Zhang et al., 2019b), we for the first time combine both *random and class-balanced sampling* strategies into CIL. Our designed network architecture $f(\boldsymbol{\theta}, \boldsymbol{\theta}_\mathrm{c}, \mathbf{x})$ for CIL-QUD contains a *shared feature extractor* $\varphi(\boldsymbol{\theta}, \mathbf{x})$, a *primary classifier* trained with class-balanced data, and an *auxiliary classifier* trained with randomly sampled data (Zhang et al., 2019b); see Fig. 3 and Sec. 3.4 for detailed illustration. In this way, the feature extractor learns from both class-balanced and randomly sampled data, where the former prevents the feature extractor from prediction preference toward majority classes, and the latter improves generalization in minority classes.

We notice that both classifiers see satisfying results, but there also exists a performance trade-off (Zhang et al., 2019b) between them. The primary classifier, trained with class-balanced data, remembers more previous knowledge and performs better on the previous/minority classes; the auxiliary classifier, trained with randomly sampled data, performs better on the new/majority classes, at the cost of forgetting previous knowledge to some extent. That trade-off motivates us to design a **c**lassifier **e**nsemble **m**echanism (CEM) for extra performance boosts. More details are referred to the supplement Section S1.1 and Algorithm 1.

### 3.4 Overall Framework of CIL-QUD

Our proposed framework, CIL-QUD, effectively integrates the above two remedies, as presented in Figure 3. First, we create Class-Balanced Batch ($\mathcal{B}_{\mathrm{CB}}$) and Random Sample Batch ($\mathcal{B}_{\mathrm{RS}}$) through class-balanced and random sampling from stored previous data and incoming new data. $\mathcal{B}_{\mathrm{CB}}$ and $\mathcal{B}_{\mathrm{RS}}$ are both used in feature extractor training, while they are separately employed to the primary and the auxiliary classifier. Second, we query the auxiliary unlabeled data by finding K-nearest-neighbors of previously stored labeled data over their feature embeddings. Details on the use of unlabeled data and feature embeddings can be found in section S2. We then randomly sample from queried unlabeled data to form an Unlabeled Data Batch ($\mathcal{B}_{\mathrm{UD}}$). In this way, we inject more "similar" knowledge of previous classes into CIL models via $\mathcal{B}_{\mathrm{UD}}$. Third, feeding the $\mathcal{B}_{\mathrm{UD}}$ together with $\mathcal{B}_{\mathrm{CB}}$ and $\mathcal{B}_{\mathrm{RS}}$, the primary classifier produces a cross-entropy loss $\mathcal{L}_{\mathrm{CB}}$ for classification and a regularization term $\mathcal{L}_{\mathrm{LwF}}$ for preventing forgetting. Similarly, the auxiliary classifier yields $\mathcal{L}_{\mathrm{RS}}$ and $\mathcal{L}_{\mathrm{LwF}}$. Note that $\mathcal{L}_{\mathrm{LwF}}$ is calculated on the unlabeled data batch, i.e, $\mathcal{B}_{\mathrm{UD}}$. In brief, CIL-QUD is cast as the following regularized optimization problem:

$$\min_{\boldsymbol{\theta}, \boldsymbol{\theta}_{\mathrm{c},1}, \boldsymbol{\theta}_{\mathrm{c},2}} \mathbb{E}_{(\mathbf{x},y) \in \mathcal{B}_{\mathrm{CB}}} \left[ \mathcal{L}_{\mathrm{CB}}(f(\boldsymbol{\theta}, \boldsymbol{\theta}_{\mathrm{c},1}, \mathbf{x}), y) \right] \quad \begin{aligned} &+ \mathbb{E}_{(\mathbf{x},y) \in \mathcal{B}_{\mathrm{RS}}} \left[ \mathcal{L}_{\mathrm{RS}}(f(\boldsymbol{\theta}, \boldsymbol{\theta}_{\mathrm{c},2}, \mathbf{x}), y) \right] \\ &+ \lambda \cdot \left[ \mathcal{L}_{\mathrm{LwF}}(\boldsymbol{\theta}, \boldsymbol{\theta}_{\mathrm{c},1}) + \mathcal{L}_{\mathrm{LwF}}(\boldsymbol{\theta}, \boldsymbol{\theta}_{\mathrm{c},2}) \right], \end{aligned} \tag{3}$$

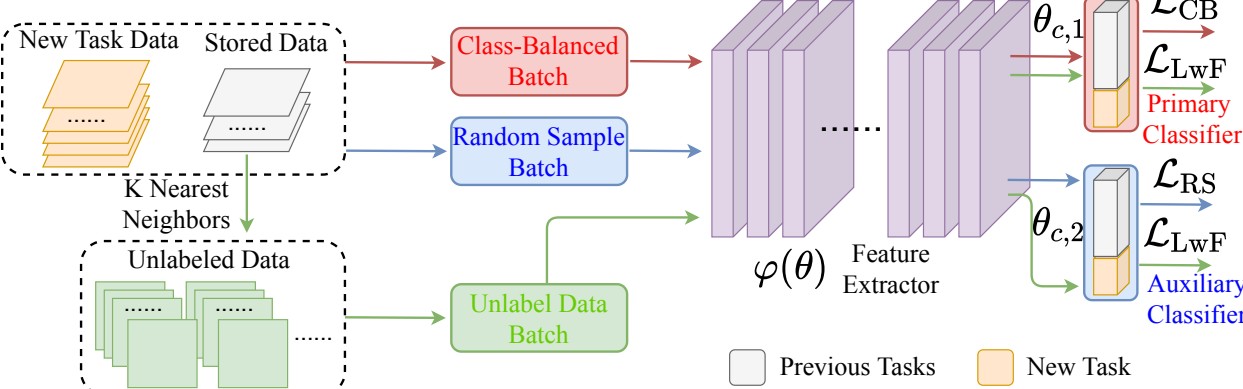

Figure 3: Overall framework of CIL-QUD. From left to right, before each training iteration, we first create three batches: Class-Balanced Batch $\mathcal{B}_{\mathrm{CB}}$ contains balanced data between stored previous classes and newly added classes; Random Sample Batch $\mathcal{B}_{\mathrm{RS}}$ includes data randomly sampled from all stored data and new task data; Unlabeled Data Batch $\mathcal{B}_{\mathrm{UD}}$ consists of auxiliary unlabeled data queried by stored previous data using the K-nearest-neighbors algorithm over the feature embeddings. Then, three batches are fed into a feature extractor $\varphi(\cdot)$, ResNet-18 (He et al., 2016). *Red arrows* (➔), *Blue arrows* (➔) and *Green arrows* (➔) represent the corresponding feed forward paths. The primary classifier takes features from $\mathcal{B}_{\mathrm{CB}}$ and $\mathcal{B}_{\mathrm{UD}}$ to calculate the objective $\mathcal{L}_{\mathrm{CB}}$ and $\mathcal{L}_{\mathrm{LwF}}$. The auxiliary classifier produces the objective $\mathcal{L}_{\mathrm{RS}}$ and $\mathcal{L}_{\mathrm{LwF}}$ with features from $\mathcal{B}_{\mathrm{RS}}$ and $\mathcal{B}_{\mathrm{UD}}$. Please zoom-in for details.

where $\boldsymbol{\theta}_{c,1}, \boldsymbol{\theta}_{c,2}$ donate the parameters from the primary and auxiliary classifiers respectively, $\lambda$ is a hyperparameter, controlling the contributions of LwF regularizers on queried unlabeled data. In our case, $\lambda = 0.5$. For tuning of hyper parameters, we perform a grid search. $\mathcal{L}_{\mathrm{LwF}}$ is selected from $\{\mathcal{KD}, \mathcal{FT}\}$ depicted in Equation 1 and 2. The auxiliary classifier serves as an implicit regularizer for preventing networks from over-fitting minority classes of previous data.

## 4 Robustified Class-Incremental Learning with Queried Unlabeled Data (RCIL-QUD)

In this section, we motivate the new setup of *robustness-aware CIL*, where a model has to maintain adversarial robustness across old and new tasks. Assisted by our unlabeled data query scheme, we propose two robustified LwF regularizers together with an add-on robust regularizer that can be plugged in CIL-QUD, leading to the RCIL-QUD.

The vulnerability of DNNs raises critical demands for improving their robustness (Goodfellow et al., 2014). However, no formal assessment of adversarial robustness has been performed in the CIL setting. It is natural to suspect that catastrophic forgetting would make learned robustness hard to sustain over new tasks too. Indeed, recent studies have identified such challenges, even in the much simpler case of transferring robustness from one source to another target domain (Hendrycks et al., 2019; Shafahi et al., 2020; Goldblum et al., 2020; Chan et al., 2019). Moreover, compared to standard generalization, DNNs need significantly more data to achieve adversarially robust generalization (Schmidt et al., 2018; Madry et al., 2018; Shafahi et al., 2020; Kurakin et al., 2016; Zhai et al., 2020), which also challenges CIL where previous classes may only have a handful of stored samples - *that is precisely **why** our queried external data can become the necessary aid and the blessing.*

To handle this new daunting setting, we first investigate the **catastrophic forgetting of robustness**. As shown in Figure 1 (b) and later in Section 5.3, neither the conventional LwF techniques nor the direct application of adversarial training (AT) (Madry et al., 2018) (the most successful defense method) can prevent the model's learned robustness from decaying over time. Here the direct application of AT refers to train over the worst-case losses penalized by standard LwF regularizers.

In our proposed RCIL-QUD, in addition to incorporating worst-case (min-max) training losses as AT, we propose to robusify our LwF regularizations with queried unlabeled data, denoted as $\mathcal{L}_{\mathrm{LwF}}^{\mathrm{R}}$, for sustaining adversarial robustness in the CIL scenario. RCIL-QUD formulation is depicted as:

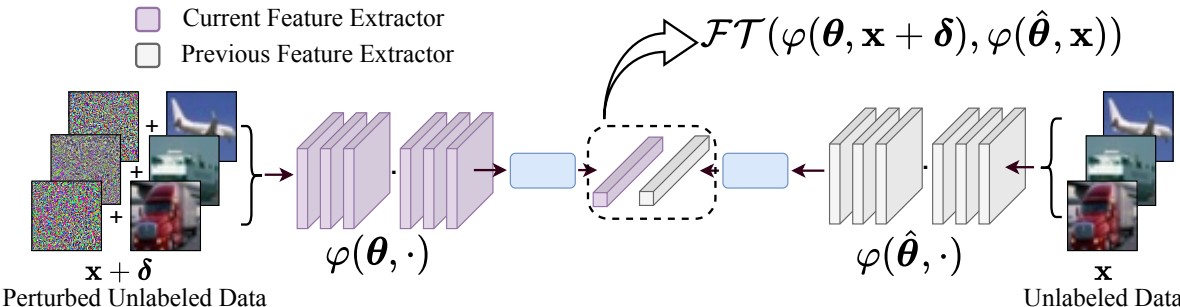

Figure 4: Illustrations of $\mathcal{RFT}$ regularization. From left to right, we first feed the perturbed unlabeled data $\mathbf{x} + \boldsymbol{\delta}$ through current feature extractor $\varphi(\boldsymbol{\theta}, \cdot)$ and get a feature vector $\varphi(\boldsymbol{\theta}, \mathbf{x} + \boldsymbol{\delta})$. Then we pass the clean unlabeled data $x$ to previous feature extractor $\varphi(\hat{\boldsymbol{\theta}}, \cdot)$ and obtain another feature vector $\varphi(\hat{\boldsymbol{\theta}}, \mathbf{x})$. $\mathcal{RFT}$ enforces the similarity between two features, where $\mathcal{FT}$ is an $\ell_1$ norm.

$$\min_{\boldsymbol{\theta}, \boldsymbol{\theta}_{c,i}, \boldsymbol{\theta}_{c,2}} \mathbb{E}_{(\mathbf{x},y) \in \mathcal{B}_{\mathrm{CB}}} \left[ \max_{\|\boldsymbol{\delta}\|_{\infty} \leq \epsilon} \mathcal{L}_{\mathrm{CB}}(f(\boldsymbol{\theta}, \boldsymbol{\theta}_{c,1}, \mathbf{x} + \boldsymbol{\delta}), y) \right] \quad + \mathbb{E}_{(\mathbf{x},y) \in \mathcal{B}_{\mathrm{RS}}} \left[ \max_{\|\boldsymbol{\delta}\|_{\infty} \leq \epsilon} \mathcal{L}_{\mathrm{RS}}(f(\boldsymbol{\theta}, \boldsymbol{\theta}_{c,2}, \mathbf{x} + \boldsymbol{\delta}), y) \right] \quad (4)$$
$$+ \gamma_1 \cdot \mathcal{L}_{\mathrm{LwF}}^{\mathrm{R}}(\boldsymbol{\theta}, \boldsymbol{\theta}_c) + \gamma_2 \cdot \mathcal{L}_{\mathcal{RTC}}(\boldsymbol{\theta}, \boldsymbol{\theta}_c),$$

where $\gamma_1, \gamma_2$ balance the effect between AT and robust regularizers. In our case, $\gamma_1 = 0.05, \gamma_2 = 0.2$. $\delta$ is the adversarial perturbation generated by Projective Gradient Descent (PGD) (Madry et al., 2018). $\epsilon$ is the upper bound of perturbations under $\ell_{\infty}$ norm. In practice, RCIL-UD can involve $\mathcal{L}_{\mathrm{LwF}}^{\mathrm{R}}$ implementations from $\{\mathcal{RKD}, \mathcal{RFT}\}$ together w. or w.o. $\mathcal{RTC}$, in Equations 5, 6 and 7 below, to pass on historic robustness.

**Robustified knowledge distillation regularizers ($\mathcal{RKD}$).** As standard LwF regularizers are found to be unsuitable for preserving robustness, we thus propose a robustified knowledge distillation term, $\mathcal{RKD}$, via introducing min-max optimization into $\mathcal{KD}$:

$$\mathbb{E}_{\mathbf{x} \in \mathcal{U}} \left[ \max_{\|\boldsymbol{\delta}\|_{\infty} \leq \epsilon} \mathcal{KD}\left(\rho(\boldsymbol{\theta}, \boldsymbol{\theta}_c, \mathbf{x} + \boldsymbol{\delta}), \rho(\hat{\boldsymbol{\theta}}, \hat{\boldsymbol{\theta}}_c, \mathbf{x})\right) \right], \quad (5)$$

where $\epsilon$, $\boldsymbol{\delta}$, $\rho$, $\boldsymbol{\theta}$, $\boldsymbol{\theta}_c$, $\hat{\boldsymbol{\theta}}$ and $\hat{\boldsymbol{\theta}}_c$ are defined similarly as equation 1 and equation 4. Our results demonstrate that $\mathcal{RKD}$ is more effective for preserving adversarial robustness on previous classes.

**Robustified feature transferring regularizers ($\mathcal{RFT}$).** We then study the robust version of feature transferring regularizer ($\mathcal{FT}$) (Shafahi et al., 2020). As mentioned recently in Shafahi et al. (2020) on robust transfer learning, the feature activations make a main source of robustness. Yet in CIL, the effectiveness of $\mathcal{FT}$ decays considerably as more tasks arrive. The limitation of $\mathcal{FT}$ motivates us to build its novel robustified version, $\mathcal{RFT}$. It enforces the similarity between the current feature representation $\varphi(\boldsymbol{\theta}, \mathbf{x} + \boldsymbol{\delta})$ of perturbed unlabeled data, and the previous representation $\varphi(\hat{\boldsymbol{\theta}}, \mathbf{x})$ of the clean unlabeled data, as shown in Figure 4. $\mathcal{RFT}$ is mathematically described as:

$$\mathbb{E}_{\mathbf{x} \in \mathcal{U}} \left[ \max_{\|\boldsymbol{\delta}\|_{\infty} \leq \epsilon} \mathcal{FT}(\varphi(\boldsymbol{\theta}, \mathbf{x} + \boldsymbol{\delta}), \varphi(\hat{\boldsymbol{\theta}}, \mathbf{x})) \right], \quad (6)$$

where $\mathcal{FT}$ has been defined in equation 2.

**Robustified TRADES regularizers on CIL ($\mathcal{RTC}$).** To further unleash the power of unlabeled data in promoting robustness, we incorporate a SOTA **r**obust defense called **T**RADES (Zhang et al., 2019a), as an add-on regularizer over $\mathcal{RKD}$ and $\mathcal{RFT}$. That is owing to the fact that TRADES does not rely on data labels, unlike the adversarial training loss (Madry et al., 2018). The formulation is:

$$\mathbb{E}_{\mathbf{x} \in \mathcal{U}} \left[ \max_{\|\boldsymbol{\delta}\|_{\infty} \leq \epsilon} \mathcal{KL}\left(\rho(\boldsymbol{\theta}, \boldsymbol{\theta}_c, \mathbf{x} + \boldsymbol{\delta}), \rho(\boldsymbol{\theta}, \boldsymbol{\theta}_c, \mathbf{x})\right) \right], \quad (7)$$

where $\rho$ is defined as above, and $\mathcal{KL}$ is Kullback–Leibler divergence. $\mathcal{RTC}$ boosts robustness via minimizing the "difference" between the predication probabilities $\rho(\boldsymbol{\theta}, \boldsymbol{\theta}_c, \mathbf{x})$ of the current model on clean samples and $\rho(\boldsymbol{\theta}, \boldsymbol{\theta}_c, \mathbf{x} + \boldsymbol{\delta})$ of the current model on adversarial samples. It is worth mentioning that $\mathcal{RTC}$ equation 7 is different from $\mathcal{RKD}$ equation 5, the former enforces the stability of predictions of the same (current) model $(\boldsymbol{\theta}, \boldsymbol{\theta}_c)$ before and after input perturbations, while the latter promotes the prediction stability of the current model at input perturbations with respect to a reference model, namely, the previously learned $(\hat{\boldsymbol{\theta}}, \hat{\boldsymbol{\theta}}_c)$.

## 5   Experiments and Analyses

### 5.1   Implementation Details

**Dataset, Memory Bank, and External Source.**   We evaluate our proposed method on CIFAR-10 and CIFAR-100 datasets (Krizhevsky et al., 2009). We randomly split the original training dataset into training and validation set with a ratio of $9:1$. We use all default data augmentations provided by Li & Hoiem (2017), and image pixels are normalized to $[0,1]$. On CIFAR-10, we divide the 10 classes into splits of 2 classes with a random order ($10/2 = 5$ tasks); On CIFAR-100, we divide 100 classes into splits of 20 classes with a random order ($100/20 = 5$ tasks). Namely, at each incremental time, the classifier dimension will increase by 2 for CIFAR-10 and 20 for CIFAR-100.

We set the memory bank to store 100 images per class for CIFAR-10 and 10 images for CIFAR-100 by default. The default external source of queried unlabeled data is 80 Million Tiny Image dataset (Torralba et al., 2008), while another source of ImageNet $32 \times 32$ (Deng et al., 2009) is investigated later. During each incremental session, we query $5,000$ and 500 unlabeled images per class for CIFAR-10 and CIFAR-100, respectively by default. Increasing the amounts of queried unlabeled data may improve performance further but incur more training costs. We use a buffer of fixed capacity to store 128 queried unlabeled images at each training iteration. More training and model selection details are referred to the supplement.

**Evaluation Metrics.**   We evaluate on ResNet18 (He et al., 2016) in terms of: (1) Standard Accuracy (SA): classification testing accuracy on the clean test dataset; and (2) Robust Accuracy (RA): classification testing accuracy on adversarial samples perturbed from the original test dataset. Adversarial samples are crafted by $n$-step Projected Gradient Descent (PGD) with perturbation magnitude $\epsilon = 8/255$ and step size $\alpha = 2/255$ for both adversarial training and evaluation, with set $n = 10$ for training and $n = 20$ for evaluation, following Madry et al. (2018).

**Existing Baselines for Comparison.**   We include several classical and competitive CIL approaches for comparison: **LwF-MC** is the multi-class implementation of Li & Hoiem (2017), and **LwF-MCMB** is its variation with a memory bank of previous data. For a fair comparison, it always keeps the same memory bank size with ours. For **iCaRL** (Rebuffi et al., 2017) and **IL2M** (Belouadah & Popescu, 2019), we use their publicly available implementations with the same memory bank sizes as ours. **IL2M** requires a second memory bank for storing previous class statistics. **DMC** (Zhang et al., 2020) is the existing CIL work that also utilizes external unlabeled data. We ensure DMC and CIL-QUD to query from the same external source.

**Variants for Our Proposed Methods.**   We first consider the vanilla version of CIL-QUD with neither queried unlabeled data nor the auxiliary classifier as Baseline among our proposals. It is still equipped with LwF regularizers yet only applied on stored labeled data[3] from previous learned tasks. We next check more self-ablations:

*a. CIL-QUD Variants:* We study **four** variants of CIL-QUD explained as follows: i) Baseline + Auxiliary classifiers: it equals to CIL-QUD without involving the usage of unlabeled data; ii) CIL-QUD w. $\mathcal{FT}$; iii) CIL-QUD w. $\mathcal{KD}$; iv) CIL-QUD w. $\mathcal{KD}$ + CEM is the variant approach equipped with classifiers ensemble mechanism (CEM).

*b. RCIL-QUD Variants:* We investigate **ten** variants of RCIL-QUD with different robust LwF regularizations on queried unlabeled data, as listed in Table 2. Classifier ensemble mechanism (CEM) is further applied to the top-performing settings. For all variants of CIL-QUD and RCIL-QUD without CEM, we evaluate the performance of the primary classifier by default, while the auxiliary classifier results are included in the supplement S3.2.

**Privacy Issues of the Unlabeled Data Collection.**   We believe that potential privacy concerns can be easily circumvented like by only querying from authorized and public datasets. To further inject privacy protection into our method, possible solutions include querying images with filtering of sensitive and offensive samples, or using generative replay: those are certainly feasible and can be our future works.

**Ethical Issues of 80 Million Tiny Image Dataset.**   We clarify that most of the experiments are completed in Spring 2020 before the withdrawal (June 2020) of 80 Million Tiny Image dataset.

---

[3] For a fair comparison, all settings are trained on the same amount of data in each training iteration. It means that the baseline approach can store extra historical data by taking over the "budget" for queried unlabeled data.

Moreover, as demonstrated in Table 4, our approach can achieve a similar performance when it queries unlabeled data from 80 Million Tiny Image or ImageNet datasets. This provides an alternative and alleviates the potential ethical issues of adopting our methods for future researchers.

---

**Takeaways:** Based on the results from Tables 1 and 2, we observe: (1) in terms of the standard accuracy, CIL-QUD w. $\mathcal{KD}$ + CEM establishes the SOTA performance; (2) in terms of the robust accuracy, CIL-QUD w. $\mathcal{RFT} + \mathcal{RTC}$ + CEM reaches a superior performance.

---

## 5.2 Improved Generalization with CIL-QUD

The standard accuracies (SAs) of all CIL models are collected in this section. As shown in Table 1, several observations could be drawn: (1) By incrementally adding our proposed components to the vanilla baseline, we find all of them contribute to preserving SA. Among the auxiliary classifier architecture (+2.37% SA), unlabeled data regularization $\mathcal{L}_{\text{LwF}}$ (+4.52% SA from $\mathcal{FT}$; +5.98% SA from $\mathcal{KD}$), and CEM (+2.65%), $\mathcal{L}_{\text{LwF}}$ is the dominant contributor, which demonstrates that queried external unlabeled data gently balances the asymmetric information between previous classes and newly added classes in CIL. (2) Comparing CIL-QUD w. $\mathcal{KD}$ with CIL-QUD w. $\mathcal{FT}$, $\mathcal{KD}$ seems to yield ob-

Table 1: Final performance for each task $\mathcal{T}$ of CIL-QUD, compared with SOTAs. Note that $\text{MT}_{\text{upper}}$ and $\text{MT}_{\text{lower}}$ train the model with multi-task learning scheme using full data and a few stored data, respectively. They act as the **empirical performance upper bounds and lower bounds** for CIL-QUD.

| Methods | CIFAR-10 (SA) | | | | | |
|---|---|---|---|---|---|---|
| | $\mathcal{T}_1$ (%) | $\mathcal{T}_2$ (%) | $\mathcal{T}_3$ (%) | $\mathcal{T}_4$ (%) | $\mathcal{T}_5$ (%) | Average (%) |
| LwF-MC (Li & Hoiem, 2017) | 28.30 | 58.10 | 50.20 | 46.00 | 60.25 | 48.57 |
| LwF-MCMB (Li & Hoiem, 2017) | 76.45 | 81.45 | 69.55 | 32.30 | 44.30 | 60.81 |
| iCaRL (Rebuffi et al., 2017) | 76.45 | 79.00 | 75.70 | 50.85 | 35.55 | 63.51 |
| IL2M (Belouadah & Popescu, 2019) | 78.20 | 64.05 | 60.40 | 38.95 | 92.10 | 66.74 |
| DMC (Zhang et al., 2020) | - | - | - | - | - | 73.66 |
| $\text{MT}_{\text{lower}}$ (empirical lower bound) | 62.75 | 59.30 | 51.10 | 37.40 | 49.95 | 52.10 |
| $\text{MT}_{\text{upper}}$ (empirical uppper bound) | 95.75 | 95.75 | 91.60 | 89.10 | 92.30 | 92.90 |
| Vanilla Baseline | 78.05 | 64.05 | 59.75 | 36.75 | 92.40 | 66.20 |
| Baseline + Auxiliary Classifier | 75.05 | 71.50 | 54.25 | 52.05 | 89.10 | 68.39 |
| CIL-QUD w. $\mathcal{FT}$ | 83.10 | 80.40 | 63.50 | 75.40 | 62.15 | 72.91 |
| CIL-QUD w. $\mathcal{KD}$ | 82.05 | 77.55 | 72.05 | 66.15 | 74.05 | 74.37 |
| CIL-QUD w. $\mathcal{KD}$ + CEM | 84.40 | 79.50 | 69.40 | 66.85 | 84.95 | 77.02 |
| Methods | CIFAR-100 (SA) | | | | | |
| | $\mathcal{T}_1$ (%) | $\mathcal{T}_2$ (%) | $\mathcal{T}_3$ (%) | $\mathcal{T}_4$ (%) | $\mathcal{T}_5$ (%) | Average (%) |
| IL2M (Belouadah & Popescu, 2019) | 22.05 | 18.70 | 33.45 | 32.35 | 82.90 | 37.89 |
| DMC (Zhang et al., 2020) | 40.32 | 39.75 | 42.10 | 46.12 | 59.36 | 45.53 |
| $\text{MT}_{\text{lower}}$ (empirical lower bound) | 14.40 | 10.50 | 12.30 | 11.15 | 16.75 | 13.02 |
| $\text{MT}_{\text{upper}}$ (empirical upper bound) | 70.65 | 67.55 | 75.05 | 68.80 | 74.55 | 71.32 |
| CIL-QUD w. $\mathcal{KD}$ | 30.85 | 39.30 | 52.65 | 41.80 | 67.20 | 46.36 |
| CIL-QUD w. $\mathcal{KD}$ + CEM | 51.95 | 46.15 | 52.80 | 38.60 | 44.10 | 46.72 |

tains a larger SA boost than $\mathcal{FT}$. (3) All results in Table 1 show CIL-QUD significantly outperforms previous SOTA methods by substantial margins on both CIFAR-10 (+10.28% SA) and CIFAR-100 (at least +1.19% SA). It demonstrates that, aided by query unlabeled data, CIL-QUD heavily reduces the prediction bias towards either the previous or the new classes. Note that our proposal also surpasses DMC (Zhang et al., 2020) by 1.19% SA on CIFAR-100, with the same amount of unlabeled data: that supplies evidence that anchor-based query is more effective than blindly sampling. In addition, we report the average performance variation when incrementally training CIL models in Figure 5. We observe that CIL-QUD w. $\mathcal{KD}$ stably does better in preserving previous knowledge besides superior final performance.

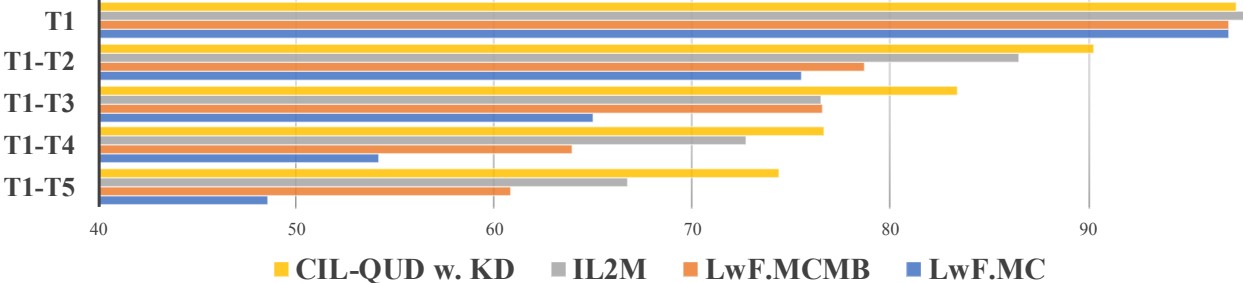

Figure 5: Average performance variation when training CIL models incrementally over 5 tasks on the CIFAR-10 dataset with selected approaches. $\text{T}_1$-$\text{T}_i$ means the model have been trained with tasks $1, \cdots, i$ ($i \in \{1, 2, 3, 4, 5\}$) continually.

## 5.3 Improved Robustness with RCIL-QUD

Under the new robustness-aware CIL scheme, the robust accuracies (RAs) of all models are collected in this section. As shown in Table 2 and S8, a number of observations can be drawn consistently on CIFAR-10 and CIFAR-100 datasets.

First, $\mathcal{FT}$ regularizer surpasses $\mathcal{KD}$ by 12.13% SA and 4.3% RA on CIFAR-10, which supports Shafahi et al. (2020); Boopathy et al. (2020)'s claim that regularizing feature presentations are crucial for robustness. Compared to our empirical performance lower bound by MTAT$_{lower}$ (explained in Table 2 caption), $\mathcal{KD}$ even hurts RA. A possible explanation is that ill-conditioned probabilities may obstacle preserving robustness. Second, robustified regulariz-

Table 2: Robust accuracy for each task $\mathcal{T}_i$ of RCIL-QUD and its variants. MTAT$_{upper}$ and MTAT$_{lower}$ adversarially train the model with multi-task learning scheme using full data and a few stored data, respectively. They act as the **empirical performance upper bounds and lower bounds** for RCIL-QUD. Here we focus on reporting RAs, while their corresponding SAs are in Table S8.

| Methods | CIFAR-10 (RA) | | | | | |
| --- | --- | --- | --- | --- | --- | --- |
| | $\mathcal{T}_1$ (%) | $\mathcal{T}_2$ (%) | $\mathcal{T}_3$ (%) | $\mathcal{T}_4$ (%) | $\mathcal{T}_5$ (%) | Average (%) |
| MTAT$_{lower}$ | 22.18 | 15.99 | 18.42 | 4.29 | 7.66 | 13.71 |
| MTAT$_{upper}$ | 54.00 | 58.70 | 39.60 | 28.30 | 21.35 | 40.39 |
| RCIL-QUD w. $\mathcal{KD}$ | 4.10 | 14.30 | 6.35 | 5.10 | 24.20 | 10.81 |
| RCIL-QUD w. $\mathcal{RKD}$ | 21.00 | 37.90 | 29.25 | 3.90 | 16.30 | 21.67 |
| RCIL-QUD w. $\mathcal{RKD}$ + CEM | 34.70 | 45.10 | 25.50 | 11.60 | 36.95 | 30.77 |
| RCIL-QUD w. $\mathcal{RKD}$ + $\mathcal{RTC}$ | 37.00 | 30.00 | 30.60 | 16.60 | 5.10 | 23.86 |
| RCIL-QUD w. $\mathcal{RKD}$ + $\mathcal{RTC}$ + CEM | 48.15 | 50.85 | 39.40 | 19.60 | 42.75 | 40.15 |
| RCIL-QUD w. $\mathcal{FT}$ | 21.70 | 26.65 | 11.90 | 6.45 | 8.85 | 15.11 |
| RCIL-QUD w. $\mathcal{RFT}$ | 25.45 | 34.05 | 19.90 | 5.15 | 6.05 | 18.12 |
| RCIL-QUD w. $\mathcal{RFT}$ + CEM | 26.75 | 35.10 | 21.50 | 6.85 | 23.80 | 22.80 |
| RCIL-QUD w. $\mathcal{RFT}$ + $\mathcal{RTC}$ | 32.45 | 29.55 | 28.10 | 23.12 | 10.13 | 24.67 |
| RCIL-QUD w. $\mathcal{RFT}$ + $\mathcal{RTC}$ + CEM | 41.45 | 37.20 | 31.95 | 16.35 | 24.30 | 30.25 |

| Methods | CIFAR-100 (RA) | | | | | |
| --- | --- | --- | --- | --- | --- | --- |
| | $\mathcal{T}_1$ (%) | $\mathcal{T}_2$ (%) | $\mathcal{T}_3$ (%) | $\mathcal{T}_4$ (%) | $\mathcal{T}_5$ (%) | Average (%) |
| MTAT$_{lower}$ | 3.65 | 2.25 | 1.85 | 1.75 | 5.30 | 2.96 |
| MTAT$_{upper}$ | 23.40 | 16.35 | 20.40 | 19.20 | 29.05 | 21.68 |
| RCIL-QUD w. $\mathcal{RKD}$ | 9.50 | 5.05 | 10.10 | 13.00 | 24.50 | 12.43 |
| RCIL-QUD w. $\mathcal{RKD}$ +CEM | 15.10 | 8.60 | 13.20 | 15.55 | 21.45 | 14.78 |
| RCIL-QUD w. $\mathcal{RFT}$ | 5.00 | 3.45 | 6.30 | 7.65 | 26.90 | 9.86 |
| RCIL-QUD w. $\mathcal{RFT}$ + CEM | 12.05 | 7.05 | 10.95 | 10.15 | 19.70 | 11.98 |

ers on unlabeled data, $\mathcal{RKD}$ and $\mathcal{RFT}$, significantly increase RA compare with their standard versions, especially for $\mathcal{RKD}$ (+11.03% SA and +10.86% RA on CIFAR-10). It suggests that injecting adversarial signals into regularizations effectively improves CIL models' resilience. Third, $\mathcal{RTC}$ further pushes RA higher, with slight TA degradation. As the red RA number shows in Table 2, it only has a negligible gap (0.24%) to the empirical RA upper bound from MTAT$_{upper}$, which we consider as an almost insurmountable milestone for achievable robustness in CIL. Lastly, the classifier ensemble mechanism (CEM) continues to benefit RAs on both CIFAR-10 and CIFAR-100.

## 5.4 Ablation Study

**Do we indeed need query, and would random unlabeled data also work?** CIL-QUD has already outperformed DMC (Zhang et al., 2020) in Section 4.2, which endorses the query way for better SAs. We further conduct experiments to justify our query also benefits RAs in RCIL-QUD w. $\mathcal{RFT}$ + $\mathcal{RTC}$. We first observe that leveraging queried unlabeled data leads to better generalization and robustness than just storing a few historical data from past tasks (i.e., Baseline in Figure 6). Then, we systematically compare the effectiveness of the following methods on utilizing unlabeled data: (i) Feature KNN: finding K nearest-neighbors for each stored anchor over feature embeddings; which is our default way; (ii) Largest Logit: finding unlabeled data with top K largest soft logits for each class; (iii) Random Pick: randomly sampling unlabeled data.

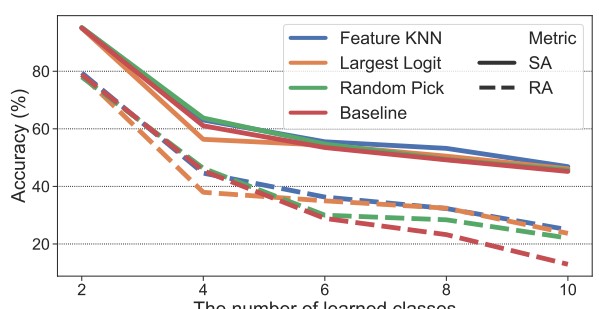

Figure 6: Average performance of RCIL-QUD w. $\mathcal{RFT}$ + $\mathcal{RTC}$ on CIFAR-10 with different unlabeled data query methods. Both RAs and corresponding SAs are reported.

As shown in Figure 6, all three options utilizing unlabeled data steadily outperform the vanilla baseline, especially in terms of RA. Our default query mechanism with feature KNN performs the best.

Table 3: Investigation of diverse sizes of the memory bank $\mathcal{S}$.

| Size per class | CIL-QUD (Standard Accuracy) on CIFAR-10 | | | | | |
|---|---|---|---|---|---|---|
| | $\mathcal{T}_1$ (%) | $\mathcal{T}_2$ (%) | $\mathcal{T}_3$ (%) | $\mathcal{T}_4$ (%) | $\mathcal{T}_5$ (%) | Average (%) |
| 10 | 65.15 | 63.20 | 57.55 | 56.70 | 67.70 | 62.06 |
| 20 | 74.90 | 72.45 | 61.30 | 71.20 | 61.40 | 68.25 |
| 50 | 82.30 | 71.75 | 66.35 | 63.65 | 71.10 | 71.03 |
| 100 | 82.05 | 77.55 | 72.05 | 66.15 | 74.05 | 74.37 |

| Size per calss | RCIL-QUD (Robust Accuracy) on CIFAR-10 | | | | | |
|---|---|---|---|---|---|---|
| | $\mathcal{T}_1$ (%) | $\mathcal{T}_2$ (%) | $\mathcal{T}_3$ (%) | $\mathcal{T}_4$ (%) | $\mathcal{T}_5$ (%) | Average (%) |
| 10 | 24.16 | 27.64 | 16.98 | 11.40 | 6.32 | 17.30 |
| 20 | 28.95 | 27.90 | 23.30 | 20.65 | 8.65 | 21.89 |
| 50 | 31.35 | 28.15 | 26.20 | 21.05 | 9.44 | 23.24 |
| 100 | 32.45 | 29.55 | 28.10 | 23.12 | 10.13 | 24.67 |

Table 4: Query unlabeled data from different sources.

| Sources | CIL-QUD (Standard Accuracy) on CIFAR-10 | | | | | |
|---|---|---|---|---|---|---|
| | $\mathcal{T}_1$ (%) | $\mathcal{T}_2$ (%) | $\mathcal{T}_3$ (%) | $\mathcal{T}_4$ (%) | $\mathcal{T}_5$ (%) | Average (%) |
| 80 Million | 82.05 | 77.55 | 72.05 | 66.15 | 74.05 | 74.37 |
| ImageNet | 78.25 | 72.45 | 68.20 | 73.80 | 70.15 | 72.57 |

| Sources | RCIL-QUD (Robust Accuracy) on CIFAR-10 | | | | | |
|---|---|---|---|---|---|---|
| | $\mathcal{T}_1$ (%) | $\mathcal{T}_2$ (%) | $\mathcal{T}_3$ (%) | $\mathcal{T}_4$ (%) | $\mathcal{T}_5$ (%) | Average (%) |
| 80 Million | 32.45 | 29.55 | 28.10 | 23.12 | 10.13 | 24.67 |
| ImageNet | 34.35 | 27.90 | 26.03 | 18.75 | 9.37 | 23.28 |

**The influence of "anchors" for query.** We adopt two representative activate learning methods to choose the "anchors": (1) maximum entropy sampling (Lewis & Gale, 1994; Settles, 2009) termed as "Entropy"; and (2) core-set selection via proxy (Coleman et al., 2020) named as "SVP". Experiments of CIL-QUD w. $\mathcal{KD}$ on CIFAR-10 have been carried out, with the settings from Table 1. We obtain: Random v.s. Entropy v.s. SVP = 74.37% v.s. 73.91% v.s. 74.76%. Results imply that different choices of "anchors" have limited effects on the achievable performance of our methods.

**Changing the size of memory bank.** Indicated by Rebuffi et al. (2017); Belouadah & Popescu (2019), the size of memory back $S$ plays a key role in the CIL performance. We conduct an ablation of CIL-QUD w. $\mathcal{KD}$ and RCIL-QUD w. $\mathcal{RFT} + \mathcal{RTC}$ with $10, 20, 50, 100$ samples stored per class, as shown in Table 3. Note that we keep the number of queried unlabeled data the same default, across all settings for a fair comparison. We observe that CIL-QUD/RCIL-QUD equipped with a large memory banks consistency achieve the superior performance, in term of both generalization and robustness. That is understandable that more anchors ensure more relevant queried data. Meanwhile, we are also encouraged to see CIL-QUD and RCIL-QUD already have competitive performance when each class has as small as 20-50 stored samples, demonstrating a nice sample and memory efficiency of our methods.

**Switching to different sources of unlabeled data** Since the 80 Million Tiny Image set is often considered to be from the same distribution with CIFAR-10/-100, we switch the unlabeled data source to ImageNet ($32 \times 32$ downsampled versions), whose images are semantically more complex and noisy. Results are presented in Table 4. We observe that the achieved performance of CIL-QUD/RCIL-QUD stay robust under a different unlabeled data distribution.

# 6 Conclusions

We introduce unlabeled data queries and show that it effectively overcome the catastrophic forgetting in the standard CIL scheme, with the help of LwF regularizations. Through anchor-based query, unlabeled data brings in more relevant information to previous classes while alleviating the information asymmetry. The power of queried unlabeled data can further extend to preserving robustness in CIL, opening up a new dimension of robustified CIL and establishing strong milestone results. Our results exemplify a significant reduction of the performance gap between incremental and non-incremental learning, in terms of both SA and RA. Similarly to other deep learning frontiers, unlabeled data evidently reveals the new promise for CIL too, if leveraged appropriately. We hope that our findings can broadly inspire people to dig deeper into utilizing unlabeled data for lifelong learning, knowledge transfer, and sustained robustness.

# Acknowledgement

Z.W. is in part supported by the U.S. Army Research Laboratory Cooperative Research Agreement W911NF17-2-0196 (IOBT REIGN), and an IBM faculty research award.

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
