# OpenReview forum: "Queried Unlabeled Data Improves and Robustifies Class-Incremental Learning"
_TMLR — Accepted by TMLR_

### Review · Reviewer_3wop · 2022-04-29

**Summary Of Contributions:**

The paper proposed a novel framework in class-incremental learning where additional unlabeled data are used by querying a small set of pre-stored past training data without the need of storing a large amount of past data. Specifically, the small set of past data, new coming data, and unlabeled data are effectively used in the proposed framework based on learning-without-forgetting regularizers and class-balance training. The adversarial robustness of the proposed method is also considered and explored through experiments on CIFAR-10/-100.

**Broader Impact Concerns:**

I don't have any concerns about the ethical implications of the work.

**Requested Changes:**

1. I suggest the authors check and update the references since many arxiv papers presented in the references are already published, such as Zhang et al. (2019c) and Chen et al. (2020b).
2. It would be better if the paper could include some discussions of recent related work in Section 2, i.e., related work in the recent two years. For example, Wang et al. ORDisCo: Effective and Efficient Usage of Incremental Unlabeled Data for Semi-supervised Continual Learning. In CVPR 2021.
3. Fix a typo in the caption of Figure 1: Summery should be Summary.

**Strengths And Weaknesses:**

Strengths:
1. The proposed class-incremental learning method is clear and straightforward. The adversarial robustness of the proposed method is also explored in this paper.
2. The experimental results of the proposed method are quite good on CIFAR-10/-100 when compared with previous methods. The authors also conducted interesting ablation studies. For example, different query mechanisms, size of the memory bank, and sources of unlabeled data are examined experimentally.
3. The paper is well-written and structured. The reading experience of this paper is enjoyable.

Weaknesses:
1. Using the stored data to query similar unlabeled data may cause large costs in computational resources and time, which makes it hard to be applied in real-time settings in practice.
2. The related work presented in the paper is quite old. All references are in/before the year 2020.

Some additional comments:
1. I'm wondering why the result of the DMC method (Zhang et al., 2019c) on CIFAR-10 is not presented in the paper, only presenting it on CIFAR-100? It is an important baseline to be compared with the proposed method.
2. The paper mentioned that the primary classifier and auxiliary classifier remember knowledge in different aspects so a classifiers ensemble mechanism (CEM) is applied for extra performance boosts. But from the experimental results, it seems to add CEM performs well than not adding it on CIFAR-10, but not on CIFAR-100. Compared with not adding CEM, adding it even makes a large performance drop (67.2 to 44.1) on one setting of CIFAR-100. So does it mean the performance of adding CEM is quite random (not sure whether it would help)?

---

> ### Author Response · Authors · 2022-05-21
> **Response to Reviewer 3wop [Cons 1-5]**
>
> Thanks for rating our proposal as novel, our empirical studies as interesting, and our paper as well-written and structured. We found the perceptive reference issues you raised very constructive and extremely helpful. We point-wisely address all your concerns as below:
>
> **[Cons 1. Large cost in querying unlabeled data?]** We respectfully argue our approach is realistically applicable. The reasons lie in:
>
> <Only query a few images per iteration.> As indicated in Section 5.1, at each training iteration, we use a buffer of fixed capacity to store 128 unlabeled images dynamically queried from the cloud or web, which are immediately discarded after this mini-batch. Thus, the additional storage and computation overhead caused by unlabeled data are minimal at any time.
>
> Meanwhile,  our idea can in fact shrink the storage demand of classic reply methods (therefore more practical), since adaptively queried “free” unlabeled data can reduce the labeled data amount needed to store, without compromising performance as shown in Sections 5.3 and 5.4.
>
> <Random query also achieves satisfying performance.> As demonstrated in Figure 6 of Section 5.4, We see our methods with random queries also reach satisfactory accuracy. This allows us to further trim down the cost during the query, in resource-limited scenarios.
>
> **[Cons 2. References are old; Update citation formats; Need more recent related works.]** Thanks for pointing out. We will thoroughly revise citation formats of arxiv papers in our revision, referring to the according conferences and journals. Meantime, we will include more discussion about recent related works, for example:
>
> (R1) ORDisCo: Effective and Efficient Usage of Incremental Unlabeled Data for Semi-supervised Continual Learning (CVPR 2021)
>
> (R2) Unsupervised Continual Learning Via Pseudo Labels (IJACI Workshop 2021)
>
> (R3) Beyond Simple Meta-Learning: Multi-Purpose Models for Multi-Domain, Active and Continual Few-Shot Learning (TPAMI Special Issues 2022)
>
> (R4) Long Live the Lottery: The Existence of Winning Tickets in Lifelong Learning (ICLR 2021)
>
> (R5) Recent Advances of Continual Learning in Computer Vision: An Overview (ArXiv 2021)
>
> A paragraph like the following will be added in our revision:
>
> “Recent investigations [R1,R2] also explore other alternative possibilities of leveraging unlabelled data to boost continual learning. For example, [R1] replays unlabeled data sampled from a conditional generator, and utilizes a consistency regularization to learn an improved continual classifier. [R2] studies continual learning in a fully unsupervised mode by assigning unlabeled data with clustered pseudo labels. Meanwhile, [R3] and [R4] enable few-shot and efficient continual learning respectively, with the assistance of unlabeled data. A recent survey paper [R5] also provides a good summary of current achievements in this field. ”
>
> **[Cons 3. DMC results on CIFAR-100.]** Thanks for pointing out. We conduct DMC on CIFAR-10 with the same configuration as Table 1. It achieves 73.66% average accuracy over five incremental tasks, which has a 3.36% lower performance compared to our best CIL-QUD variant (i.e., 77.02%). We will include these additional results in our revision.
>
> **[Cons 4. Apply CEM on CIFAR-100 hurts?]** Great Gatch. We politely argue that the CEM is beneficial on CIFAR-100 overall. As shown in Table 1, after applying CEM on CIFAR-100, (i) the average accuracy of five tasks is improved from 46.36% to 46.72%; (ii) the accuracy of task 1 is boosted from 30.85% to 51.95%; (iii) the accuracy of task 2 is enhanced from 39.30% to 46.15%; (iv) the accuracy of task 3 is improved 52.65% to 52.80%. Actually, we observe that CEM leads to a more “balanced” prediction among five tasks, where the performance of the first few tasks is substantially improved at the cost of some accuracy drops in the last tasks. Note that in average performance, CAM is consistently helpful in both CIFAR-10 and CIFAR-100.
>
> The rationale of CEM is to select previous task-specific (sub-)classifiers (more “balanced”) to perform prediction based on the voted “predicted task ID”. Therefore, our design philosophy (Section 3.1 and S1.1) is matched with our results in Table 1 and Table 2.
>
> **[Cons 5. Typos.]** Many thanks for such detailed comments. We will address all typos in our revision.

---

> > ### Comment · Reviewer_3wop · 2022-05-23
> > **Post-rebuttal comments**
> >
> > Thank the authors very much for the rebuttal! The authors have addressed most of my concerns (e.g., about the practical usage of the proposed method and the experimental results). Considering the revised version that adds more discussions of recent work and updates the references, I would tend to accept this paper.

---

### Review · Reviewer_VL3x · 2022-05-02

**Summary Of Contributions:**

This paper proposes an algorithm and studies some aspects of class-incremental learning. The paper has a few contributions. The first contribution is to utilize ideas from class-imbalanced learning, since if we allow to keep some training data from previous tasks (classes), this will lead to an imbalanced class learning setup with a lot of samples for the current task/classes and a few samples for the old task/classes. The next contribution is to use unlabeled data for class-incremental learning. Although Zhang et al. 2019c also used unlabeled data for class-incremental learning, this paper chooses similar ones to the stored training data (from old tasks) instead of using all of the unlabeled data, which seems to be effective in experiments and the burden of training will be lower. Finally, the paper studies the adversarial robustness of class-incremental learning and proposes an robustified version of the proposed method.

**Broader Impact Concerns:**

The paper does not have a broader impact statement. However, I do not see any immediate concerns for future negative impact to our society. I wrote my comments for potential ethical isuses in the prevous section (Strengths and Weaknesses).

**Requested Changes:**

I wrote my main comments/suggestions to improve the paper in the previous section.

The important ones in my opinion are "ethical issues", "experiments", and "motivation of the paper". The "story of the paper" is less important.

Some other minor comments are:
- Figure 1's caption: summery is summary?
- Page 2, Section 1.1: reply is replay?

**Strengths And Weaknesses:**

Strengths:

Borrowing ideas from class-imbalanced learning to tackle class-incremental learning is interesting and may be helpful for future CIL work, especially when we are allowed to have a small set of previous data from old classes. The paper also utilize unlabeled data by querying them by taking the nearest neighbors of the training data of previous tasks (CIL-QUD). In the latter sections, the paper studies adversarial robustness in CIL and proposes a robustified version of their CIL-QUD.

Weaknesses:

Motivation of the paper -- Zhang et al. 2019c's motivation for utilizing unlabeled data came from how storing past training data is unrealistic in the CIL problem. In the paper under review, we use both past training data and a large set of unlabeled data. Since this makes the problem setting easier (we have access to both), the paper further needs to motivate if this is a realistic scenario.

Ethical issues -- The experiments rely on 80 million tiny image dataset, but in 2020, the authors have withdrawn the dataset due to derogatory terms as categories and offensive images. See https://groups.csail.mit.edu/vision/TinyImages/ for the details. Although I am not an ethics expert, it seems problematic to continue using a withdrawn dataset with ethical issues.

Experiments -- The experiments are conducted with only 5 tasks, while most class-incremental learning papers perform experiments with much more.  For example, Zhang et al. 2019c performs incrementally learning 5, 10, 20, and 50 classes at a time, with iCIFAR-100. It would be more informative to see if the proposed method can maintain accuracy with respect to a higher number of tasks.

Story of the paper -- Although the paper has several contributions for the class incremental learning problem, the contributions seem to be independent from each other and it was hard to grasp the main/core contribution when I first read the paper. For example, the discussions and experiments that aim to investigate robustness is independent from the former sections, and the contribution of balanced learning is independent from using unlabeled data. In my opinion, the class-balanced/random sample component in the proposed algorithm is the most interesting contribution of the paper, but the paper does not emphasize this at all (e.g., the paper does not mention it in the abstract and the conclusion).

---

> ### Author Response · Authors · 2022-05-21
> **Response to Reviewer VL3x [Cons 1-5]**
>
> Thanks for rating our paper as effective, interesting, and helpful. We point-wisely address all your concerns as below:
>
> **[Cons 1. Need to motivate if it is a realistic setting.]** We believe our setting to be very realistic and meaningful. Overall, our method is the same broadly applicable as classical replay such as iCal/GEM, and can perform better.
> In fact, this research is directly motivated by the author team’s current project of networked robots: a continual learning agent with a low memory footprint but allowing for internet communication. In this real-world scenario, we do plan for real prototyping/production of our method.
>
> Highlights of our settings:
>
> (1) there is almost no extra “storage cost of the unlabeled dataset”. Since at each training iteration, we use a buffer of fixed capacity to store 128 unlabeled images dynamically queried from the cloud or web, which are immediately discarded after this mini-batch (Section 5.1). Thus, while labeled anchors are held locally, the additional storage overhead caused by unlabeled data is minimal at any time.
>
> (2) Our idea can in fact shrink the storage demand of classic reply methods (therefore more practical), since adaptively queried “free” unlabeled data can reduce the labeled data amount needed to store, without compromising performance as shown in Section 5.3 and 5.4.
>
> **[Cons 2. Ethical issues of 80 million tiny image dataset.]** Many thanks for pointing out. We want to clarify that most of the experiments are completed in Spring 2020 before the withdrawal (June 2020) of 80 million tiny image dataset. We will include this clarification in our revision.
>
> Moreover, as demonstrated in Table 4, our approach can achieve a similar performance when it queries unlabeled data from 80 million tiny images or ImageNet datasets. This provides an alternative and alleviates the potential ethical issues of adopting our methods for future researchers.
>
> Lastly, we also notice that 80 million tiny image dataset is still used in recent publications. For instance, table 1 of [R1] uses models trained from 80 million tiny image dataset.
>
> [R1] Automated Discovery of Adaptive Attacks on Adversarial Defenses [NeurIPS 2021]
>
> **[Cons 3. Experiments with more tasks.]** Great suggestion. We follow reviewer VL3x’s comments and conduct additional experiments on CIFAR-100 by incrementally learning 10 classes at a time (i.e., a higher task number of 10). Results are collected in the below table, where settings are the same as the ones in Table 1. We see our proposed methods maintain superior performance (at least 7.91% accuracy gains), compared to previous approaches. These extra results and related discussions will be added to our revision. Meanwhile, more experiments like continual learning with 20 tasks are promised in our final verison.
>
> |Accuracy (%)|T_1|T_2|T_3|T_4|T_5|T_6|T_7|T_8|T_9|T_10|**Average**|
>
> |iCaRL|5.90|7.50|4.50|2.80|9.00|8.00|28.20|38.50|59.60|80.20|**24.42**|
>
> |IL2M|19.90|24.10|19.80|12.90|21.30|21.70|29.90|34.80|40.30|89.80|**31.45**|
>
> |Baseline+Auxiliary Classifier (Ours)|21.20|32.10|23.00|22.70|21.70|31.70|39.60|33.80|40.30|54.30|**32.02**|
>
> |CIL-QUD w. KD (Ours)|29.04|33.94|32.54|27.94|32.74|29.64|47.94|45.34|47.24|67.24|**39.36**|
>
> **[Cons 4. Loosely connected story?]** We respectfully disagree.
>
> <Connection with robustness.> We see our two parts of the standard generalization and robustness of CIL as tightly connected. It is known that standard and robust accuracies can have inherent tension, but the latter was rarely formally examined in CIL. Note that CIL-QUD and RCIL-QUD share most of their pipelines in common - only the latter “robustified” some components using an adversarial training idea, thus we have one coherent methodology (querying unlabeled data) throughout the paper, and we show it can “kill two birds”.
>
> <Contribution of class-balanced/random samplings.> Many thanks for acknowledging our algorithm as interesting. We will follow reviewer VL3x’s suggestion and mention these parts in the abstract and conclusion as contributions. But our key point is still: Querying unlabeled data to improve the standard and robust generalization of CIL models, through appropriate learning-without-forgetting (LwF) regularizers and class-balance training.
>
> **[Cons 5. Minors.]** Yes. They are typos. We will address them in our revision and sincerely appreciate detailed comments.

---

> > ### Comment · Reviewer_VL3x · 2022-05-23
> > **Thank you for the response**
> >
> > Thank you for answering my questions and concerns. I will submit my final review soon based on these discussions.

---

> > ### Comment · Action_Editors · 2022-05-26
> > **Concerning the 80 million tiny image dataset**
> >
> > I don't see the 3rd reason valid in this argument. A lot of recent publications in 3 top ML conferences are not reproducible as you-know-why which cannot be a reason for others to submit irreproducible papers. For this kind of ethical issues, we just need to consider whether it is ethically good to do so --- if yes, just go on, and if no, definitely stop, regardless of what others do. I have to emphasize this point since the paper is open review and the 3rd reason is quite misleading.
> >
> > Given that the 1st and 2nd reasons look valid, if the authors don't want to hide the experiments based on the possibly problematic 80 million tiny image dataset, I think the authors can simply encourage people to query from ImageNet for temporary unlabeled data or even make querying from ImageNet the only choice in the released code to prevent people from using the 80 million tiny image dataset.

---

### Review · Reviewer_vEWF · 2022-05-05

**Summary Of Contributions:**

This paper proposes a continual learning framework boosted by adaptively queried unlabeled training data (namely CIL-QUD) to mitigate catastrophic forgetting and thus enhance natural generalization. Further, the authors propose a robust version of CIL-QUD (namely RCIL-QUD) to prevent catastrophic forgetting of adversarial robustness and thus improve robust accuracy. The experiments validate the effectiveness of the proposed framework.

**Broader Impact Concerns:**

This paper leverages unlabeled data to boost continual learning. As the authors say in the paper, the unlabeled training data could be crawled from the Internet. Therefore, this would cause an invasion of personal privacy due to the unauthorized collection of the data.

**Requested Changes:**

- Please give some description and experimental evidence for the phenomenon called catastrophic forgetting of adversarial robustness,

- It would be better to show the results based on different models to validate the generalization capability of the proposed method on various models.

- It would be helpful to report the mean and standard variance of all the results, to remove the concern incurred by randomness.

- As the authors conduct lots of variants of CIL-QUD, I recommend giving a summary of the technique that can obtain state-of-the-art performance.


**Strengths And Weaknesses:**

**Strengths**

- The proposed technique is well motivated and clearly distinguished from prior works.

- The authors show the performance of various variants of CIL-QUD and RCIL-QUD, which shows the importance of each technical component.

- This paper discovers the catastrophic forgetting of adversarial robustness in continual learning. Further, the authors modify CIL-QUD into a min-max optimization objective along with the TRADES loss (called RCIL-QUD) to mitigate the above issue.

**Weaknesses**
- The phenomenon called catastrophic forgetting of adversarial robustness seems to be unclear to me, although the authors mention it in the Introduction section.

- All the experiments are based on only one model, i.e., ResNet18.

- The 'anchors' for each training class are randomly picked. I am concerned whether the choice of 'anchors' will influence the performance.

---

> ### Author Response · Authors · 2022-05-21
> **Response to Reviewer vEWF [Cons 1-3]**
>
> Thanks for rating our paper as well-motivated and important. We point-wisely address all your concerns as below:
>
> **[Cons 1. Clarify the catastrophic forgetting.]** Actually, it is similar to the catastrophic forgetting of standard generalization but with different evaluation metrics.
>
> <Description.> When (robust) trained on one task, then (robust) trained on a second task, models “forget” and have low adversarial robustness on the first task. In other words, robustifying models on a new task leads to robustness degradation on previous tasks. We call this behavior as the catastrophic forgetting of adversarial robustness. We will further clarify this phenomenon in our revision.
>
> <Empirical Evidence.> As shown in Figure 1 (b), we perform adversarial training to incrementally robustify models on five sequential tasks. After learning the last task, we can see the adversarial robustness (robust accuracy) on the first task deteriorated to near 0.
>
> **[Cons 2. Experiments with new different models.]** Thanks for the great suggestion. We choose another classic model backbone, i.e., ResNet-32, in CIL literature [R1, R2]. New experiments are conducted on CIFAR-100 and learn 5 and 10 tasks in an incremental manner. Other settings are the same as the one in Table 1. We obtain:
>
> Incrementally learning 5 task: DMC v.s. Our CIL-QUD w. KD = 46.32% v.s. 48.07%
>
> Incrementally learning 10 tasks: DMC v.s. Our CIL-QUD w. KD = 35.99% v.s. 40.16%
>
> The consistent performance improvements on the new model validate the generalization capability of our proposals. More experiments about other backbones like VGG are promised in our final version.
>
> [R1] Class-incremental Learning via Deep Model Consolidation
>
> [R2] iCaRL: Incremental classifier and representation learning.
>
> **[Cons 3. The influence of “anchors”.]** As suggested by review vEWF, we adopt two representative activate learning methods to choose the “anchors”: (1) maximum entropy sampling [R3,R4] termed as “Entropy”; and (2) core-set selection via proxy [R5] named as “SVP”. Experiments of CIL-QUD w. KD on CIFAR-10 have been carried out, with the settings from Table 1. We obtain: Random v.s. Entropy v.s. SVP = 74.37% v.s. 73.91% v.s. 74.76%. Results imply that different choices of “anchors” have limited effects on the achievable performance of our methods.
>
> As shown in the ablation study of Section 5.4, we see the size of the memory bank and the query approaches have a larger influence on the performance of our methods.
>
> [R3] A sequential algorithm for training text classifiers
>
> [R4] Settles, B. Active learning.
>
> [R5] Selection via proxy: Efficient data selection for deep learning

---

> ### Author Response · Authors · 2022-05-21
> **Response to Reviewer vEWF [Cons 4-6]**
>
> **[Cons 4. Mean and standard deviation.]** Thanks for the constructive suggestion. Due to the limited time period of rebuttal, we currently only repeat CIL-QUD w. KD on CIFAR-10 (table 1) for five runs with different random seeds. The average performance and one standard deviation are 74.41% and 0.18%. We see our methods quite stable to the random seeds. More numbers about mean and standard deviation will be provided in our final version.
>
> **[Cons 5. A summary of techniques with SOTA performance.]** Great suggestion. Based on the results from Tables 1 and 2, we observe: (1) in terms of the standard accuracy, CIL-QUD w. KD + CEM establishes the SOTA performance; (2) in terms of the robust accuracy, CIL-QUD w. RFT + RTC + CEM reaches a superior performance. We will give a detailed summary of techniques with SOTA performance, in our revision.
>
> **[Cons 6. Privacy issues of the unlabeled data collection.]** Thanks for the reminder. We respectfully argue that potential privacy concerns can be easily circumvented like by only querying from authorized and public datasets. To further inject privacy protection into our method, possible solutions include querying images with filtering of sensitive and offensive samples, or using generative replay: those are certainly feasible and can be our future works. We will add the above discussion and clarification in our revision.

---

> ### Comment · Action_Editors · 2022-05-26
> **Any post rebuttal comments?**
>
> Hi Reviewer vEWF,
>
> Do you have any post rebuttal comments? We are (or I am) on a tight schedule, and I am awaiting your official recommendation (there are already two).
>
> --Gang

---

> > ### Comment · Reviewer_vEWF · 2022-05-26
> > **Thanks for authors' responses**
> >
> > Thanks for the authors’ responses.
> >
> > My concerns have been addressed.
> >
> > Therefore, I am inclined to accept this paper.
> >
> > --Reviewer vEWF

---

### Comment · Action_Editors · 2022-05-09
**Rebuttal**

Dear authors,

If possible, please first address the concerns that don't need to run additional experiments, rather than address all concerns at once in the end. Since the review process of TMLR is interactive and the available rebuttal time is quite limited, this strategy can let you fully make use of the advantages of our review process. You can basically regard us as the journal version of ICLR, so you can also do what you like to do when submitting your valuable work to ICLR.

Best,
Gang

---

> ### Comment · Action_Editors · 2022-05-20
> **Reminder**
>
> While the NeurIPS submission deadline was just passed and I can understand that you should be busy and tired with your submissions, we are also in a fairly tight schedule on the TMLR side, so please address the concerns from our reviewers **as soon as possible**. Thanks.

---

> ### Comment · Action_Editors · 2022-05-23
> **Author responses came in**
>
> Dear reviewers,
>
> The author responses are in the system now so could you take a look at your convenience?
>
> Best,\
> Gang

---

### Author Response · Authors · 2022-05-21
**General Response**

To AE and all reviewers:

We would like to thank all reviewers for providing many useful feedbacks and thank AE for kind reminders. We sincerely apologize for this delayed response, since the author team had been occupied by another deadline just passed.

Below we address all questions raised and provide point-to-point responses, **with almost all required additional experiments**. We thank all reviewers for appreciating our novel algorithms, interesting & effective empirical studies, and well-written & structured paper.

We hope our responses, although coming a bit late in this time window (we apologize again), have clarified all confusion and could help reviewers more positively assess our work. We thank all reviewers and AE’s time again.

---

> ### Author Response · Authors · 2022-05-25
> **Update of Revised Drafts**
>
> Dear AE and all reviewers,
>
> Most of the promised changes (>95%) have been included in our revised main draft and appendix, which are marked in blue. We will keep updating our draft.
>
> We sincerely appreciate all reviewers’ and ACs’ time and efforts in reviewing our paper. We truly thank all for the insightful and constructive suggestions, which helped further improve our paper. We genuinely appreciate the reviewer **3wop** for voting for acceptance.
>
> We are confident that our response should have cleared the air, and we are happy to answer any additional questions and provide more information.
>
> We really thank all reviewers’ and AEs' time and efforts again.
>
> Best wishes,
>
> Authors

---

### Decision · Action_Editors · 2022-05-29

**Recommendation:** Accept as is

**Comment:**

This is an interesting paper working on an important problem "class-incremental learning". It proposed several components to improve (wrt natural accuracy) and robustify (wrt adversarial accuracy) class-incremental learning methods. Although TMLR doesn't emphasize novelty and significance, the two factors are both fairly high to the reviewers and me. The authors did a good job in the rebuttal to address the concerns from the reviewers, and then all reviewers voted for acceptance. Furthermore, given that most of the promised changes (>95%) have already been included, I feel the paper could be accepted as is.

---

> ### Comment · Editors_In_Chief · 2022-12-08
> **Featured Certification**
>
> In later discussions, the AE proposed to award this paper a Featured Certification, which we are happy to approve. Congratulations! Here's a description written by the AE, of the reasons for giving this work a Featured Certification.
>
> _The problem studied in the paper, namely how to mitigate catastrophic forgetting when training big models with more and more new data (specifically, class-incremental learning here), is very critical for deploying learning-based models in the real world. The paper proposed a very interesting idea to reduce the storage burden of SOTA methods, that is, to prepare a big pool of unlabeled data that can represent various data distributions and then query relevant unlabeled data from the pool on-the-fly (so that whenever necessary we can fetch a lot of unlabeled data following our desired data distribution). Furthermore, the paper proposed the first strong baseline for adversarial robustness under such a problem setting, and demonstrated that catastrophic forgetting can be successfully mitigated for not only generalization to natural examples but also robustness against adversarial examples. Many researchers/practitioners working on deep learning may be potentially interested in the ideas presented in the paper, and thus I think it should deserve a Feature Certification._